# Exploration of Potential Breath Biomarkers of Chronic Kidney Disease through Thermal Desorption–Gas Chromatography/Mass Spectrometry

**DOI:** 10.3390/metabo13070837

**Published:** 2023-07-11

**Authors:** Si-Hyun Seong, Hyun Sik Kim, Yong-Moon Lee, Jae-Seok Kim, Sangwoo Park, Jieun Oh

**Affiliations:** 1Mass Spectrometry & Advanced Instrumentation Group, Korea Basic Science Institute, Cheonju 28119, Republic of Korea; sean7067@kbsi.re.kr (S.-H.S.); hskim@astams.com (H.S.K.); 2College of Pharmacy, Chungbuk National University, Cheongju 28644, Republic of Korea; ymleefn@gmail.com; 3ASTA Corporation, Research & Development Center, Suwon 16229, Republic of Korea; 4Department of Laboratory Medicine, Kangdong Sacred Heart Hospital, Hallym University College of Medicine, Seoul 05355, Republic of Korea; jaeseokcp@gmail.com; 5Koscom Fund Services Corporation, Seoul 07330, Republic of Korea; parkswpd@gmail.com; 6Department of Internal Medicine, Kangdong Sacred Heart Hospital, Hallym University College of Medicine, Seoul 05355, Republic of Korea

**Keywords:** chronic kidney disease, hemodialysis, uremic toxins, breath markers, volatile organic compounds, mass spectrometry

## Abstract

Breath volatile organic compound (VOC) analysis is a non-invasive tool for assessing health status; the compositional profile of these compounds in the breath of patients with chronic kidney disease is believed to change with decreasing renal function. We aimed to identify breath VOCs for recognizing patients with chronic kidney disease. Using thermal desorption–gas chromatography/mass spectrometry, untargeted analysis of breath markers was performed using breath samples of healthy controls (*n* = 18) versus non-dialysis (*n* = 21) and hemodialysis (*n* = 12) patients with chronic kidney disease in this cross-sectional study. A total of 303 VOCs alongside 12 clinical variables were used to determine the breath VOC profile. Metabolomic analysis revealed that age, systolic blood pressure, and fifty-eight breath VOCs differed significantly between the chronic kidney disease group (non-dialysis + hemodialysis) and healthy controls. Thirty-six VOCs and two clinical variables that showed significant associations with chronic kidney disease in the univariate analysis were further analyzed. Different spectra of breath volatile organic compounds between the control and chronic kidney disease groups were obtained. A multivariate model incorporating age, 2-methyl-pentane, and cyclohexanone showed high performance (accuracy, 86%) in identifying patients with chronic kidney disease with odds ratios of 0.18 (95% CI, 0.07–2.49, *p* = 0.013); 2.10 (0.94–2.24, *p* = 0.025); and 2.31 (0.88–2.64, *p* = 0.008), respectively. Hence, this study showed that renal dysfunction induces a characteristic profile of breath VOCs that can be used as non-invasive potential biomarkers in screening tests for CKD.

## 1. Introduction

Human breath analysis has become a promising research field that has recently attracted significant interest owing to advances in analytical techniques. Exhaled human breath typically consists of nitrogen (78%), oxygen (16%), carbon dioxide (4–5%), hydrogen (5%), inert gases (0.9%), and volatile organic compounds (VOCs) [1]. VOCs can vary depending on a person’s metabolism, which changes when someone becomes sick. For decades, humans have used the superior sense of smell of dogs to identify illicit drugs or explosives. Sniffing animals have also been trained to detect human diseases, such as cancers, tuberculosis, or even coronavirus disease 2019 [2,3]. In all cases, animals are presumed to detect chemicals emitted by humans through their body odors or breath. Researchers do not know exactly which components the animals smell; however, it is understood that these diseases cause the human body to release the characteristic patterns of VOCs in scents.

The kidneys remove waste products and extra water from the body and maintain homeostasis for the normal functioning of cells and organs. Chronic kidney disease (CKD) is a long-term condition of gradual loss of filtering function that results in the progressive accumulation of more than one hundred uremic retention solutes [4,5]. In patients with CKD, the compositional profile of breath VOCs can change because of the limited ability of kidneys to eliminate metabolic products from the blood. Metabolites produced in the body are transported to the alveoli, and trace amounts of volatile substances are exhaled via respiration. Breath VOCs can be analyzed using mass spectrometry, and changes in the VOC profile according to health status can be used as a breathprint for human diseases. Evidence suggests that patients with CKD have higher levels of VOCs in their breath than those of healthy individuals [6,7,8,9,10]. CKD is clinically silent and asymptomatic in many cases and is often detected during health checkups or is not detected until the late stage. The severity of CKD can be quantified by a low estimated glomerular filtration rate (eGFR) and increased urinary albumin levels, both of which require blood sampling and/or urine collection [11]. Breath VOC analysis is a noninvasive tool for assessing health status information. Exhaled breath is one of the most easily collected samples and contains profound but unknown information related to human diseases. Olfactory evaluation can provide diagnostic clues and guide further evaluation.

In this study, we analyzed more than 300 VOCs in the breath of 51 participants (18 normal healthy controls, 21 patients with CKD, and 12 patients with CKD undergoing hemodialysis) using thermal desorption–gas chromatography/mass spectrometry (TD–GCMS) along with clinical variables. This dataset allowed us to test the hypothesis that the normal and CKD states can be distinguished based on breath VOCs that change as CKD progresses.

## 2. Materials and Methods

### 2.1. Study Design and Participants

To address our hypothesis, we designed a cross-sectional study to compare the compositional profile of breath VOCs between normal healthy controls and CKD patients, along with confounding clinical variables. A total of 51 participants were included in this study: 18 healthy controls, 21 non-dialysis outpatients with CKD (CKD-ND), and 12 hemodialysis patients (HD). All participants were adults aged 19 years or more, and patients with CKD-ND and HD were recruited from Kangdong Sacred Heart Hospital between March and September 2021. CKD was defined as the presence of kidney damage or decreased kidney function for three or more months, irrespective of the cause [11]. The inclusion criterion for CKD-ND was stable renal function for >3 months, without acute kidney injury or hospitalization. Hemodialysis patients underwent regular hemodialysis three times a week for more than three months and had no acute diseases. Normal healthy controls were adults who were confirmed to be free of kidney disease through a general health checkup and had CKD-EPI eGFR of ≥80 mL/min/1.73 m^2^, with no proteinuria. This study was approved by the Institutional Review Board of Kangdong Sacred Heart Hospital (IRB no. 2019-10-001) and was conducted in compliance with the Helsinki Congress and Declaration of Istanbul. Written informed consent was obtained from all participants. Laboratory data were collected to determine the biochemical parameters.

### 2.2. Breath Sample Collection and Preparation

Pre-concentration methods can detect trace concentrations of VOCs in exhaled breath samples. Sorbent-containing thermal desorption stainless steel tubes are most frequently used to adsorb and enrich VOCs in breath [12]. Prior to sample collection, each thermal desorption tube (Tenax TA, 3.5 inch, 60–80 mesh, 230–250 mg, KNR, Namyangju, Republic of Korea) was heated and sealed to ensure absence of any residual gas in the tube. Tedlar bags are difficult to fill due to their low gas flow conductance, and hence a 10 L polyester bag (Top-Trading, Seoul, Republic of Korea), which is a disposable sampling device, was used for breath collection. Exhaled breath samples were collected from participants in the morning. Breath samples from normal controls and non-dialysis CKD patients were collected once, and breath samples from hemodialysis patients were collected twice, before and after hemodialysis. Breath collection was conducted in the same outpatient clinic room of our hospital, in a clean and quiet environment, which was dedicated for breath collection on the sampling day. The participants were asked to breathe deeply and inflate a 10 L polyester bag. A Gilair Plus pump (Sensidyne, St. Petersburg, FL, USA) was used to force ambient air to flow at a rate of 250 mL/min through the desorption tubes (active sampling). The thermal desorption tube was connected and VOCs were adsorbed for 12 min from the 3 L aliquot flow from the polyester bag. Three consecutive samples were collected from each patient. To collect the background air, 3 L of air from the sampling room was concentrated in a thermal desorption tube in duplicate using a Gilair Plus pump at a pumping speed of 250 mL/min for 12 min. The VOC-adsorbed thermal desorption tube was immediately sealed and stored in an icebox to avoid deterioration of collected samples.

### 2.3. VOC Analysis Using TD–GCMS

The adsorbed VOCs were vaporized and analyzed by TD–GCMS (Turbomatrix 650, Perkin Elmer, Waltham, MA, USA; 7890B-GC and 5977A-MS, Agilent Technologies, Santa Clara, CA, USA). A fused silica capillary GC column, an Elite-5 ms column (60 m × 0.32 mm × 1 μm, Perkin Elmer, Waltham, MA, USA), was used to analyze the desorbed VOCs in the breath. Helium with chromatographic purity (220 kPa) was used as the carrier gas. The VOCs were desorbed for 15 min from the thermal desorption tube to which breath VOCs were adsorbed. Maintaining oven, cold trap, and transmission line temperatures of 250 °C, −30 to 300 °C (10 min) at 33 °C/min, and 250 °C, respectively, gas chromatography column temperature was held at 30 °C for 10 min, following by gradual ramping (3.4 °C/min to 50 °C and 5 °C/min to 100 °C with a hold for 10 min, and at 3.5 °C/min to 150 °C and at 100 °C/min to 250 °C with a hold at 250 °C for 10 min) [12,13]. The electron impact (EI) voltage in the mass spectrometer was set to 70 eV. The mass detection range varied from 30 to 350 *m*/*z*. Compound identification was performed based on the retention time in gas chromatography and ion fragmentation patterns using Wiley and NIST GC–MS libraries (Wiley Spectral Libraries, Wiley Science Solutions, Hoboken, NJ, USA, https://sciencesolutions.wiley.com/solutions/technique/gc-ms/wiley-registry-of-mass-spectral-data-12th-edition/ (accessed on 29 January 2023)) and compared with previously reported retention time data in the literature [14,15]. The relative concentration of each compound was calculated from the peak areas of the selected ion chromatograms. Most of the VOCs observed in the TD–GCMS experiment have been reported previously [13,14,15]. Untargeted metabolomics using TD–GCMS was performed on the VOCs observed in breath samples.

### 2.4. Statistical Analyses

We compared the levels of 303 breath VOCs and 12 clinical variables between the healthy control and CKD groups, which included both non-dialysis CKD and hemodialysis patients. The VOC values before hemodialysis were used to investigate the relationship between renal dysfunction and breath VOCs. For descriptive analysis, continuous variables are expressed as mean ± standard deviation (SD), and categorical variables are expressed as frequencies (percentages). Continuous variables included six clinical variables (age, body weight, height, body mass index, and systolic and diastolic blood pressure) and 303 breath VOCs. Because most of the measured VOC values ranged between 10^6^ and 10^8^, these were scaled down as follows: SQRT (breath VOC value-background room air VOC value)/1000. The categorical variables included six clinical variables: sex, diabetes, hypertension, dyslipidemia, cerebrovascular accident (CVA), and coronary heart disease (CHD).

Categorical variables were compared using a χ^2^ test between groups as appropriate. For continuous variables, we conducted a Student’s *t*-test and univariate logistic regression to select significant VOCs and clinical variables that could distinguish between the normal and CKD (non-dialysis CKD + HD) groups.

Principal component analysis (PCA) was performed with breath VOCs and clinical variables that showed *p*-values of odds ratio < 0.05 in the univariate analysis. PCA was used to determine the possibility of separating patients with CKD from healthy controls by analyzing breath VOCs.

Next, a multivariate analysis was performed in two steps. First, significant continuous variables were selected from 100 repeated training and test sets. Briefly, 51 participants were randomly sampled into the training set (67%) and the test set (33%), and a multivariate model was constructed with the training dataset; further, the accuracy was derived by applying this model to the corresponding test set. This process was repeated one hundred times and the variables used in each multivariate model were listed based on the number of times they were used. The most frequently used variables were determined as the selected continuous variables. In the second step, the final multivariate logistic regression model was built by integrating the selected continuous variables in the first step and four significant categorical variables selected using the χ^2^ test. We used stepwise regression with forward selection and backward elimination to obtain a model that minimized the Akaike information criterion (AIC) while avoiding overfitting. Receiver operating characteristic (ROC) curve analysis was performed by applying the final multivariate model to all participants. The software package R version 4.1.1 (www.r-project.org; The R Foundation for Statistical Computing, Vienna, Austria, accessed on 29 January 2023) was used for statistical analyses, and a *p*-value of <0.05 was considered significant. A flowchart of the statistical analyses is shown in Figure 1.

## 3. Results

### 3.1. Characteristics of Study Participants

All participants were confirmed eligible and included for analysis in this study. Table 1 summarizes the demographic and clinical characteristics of the study participants. The mean age of the normal, CKD-ND, and HD groups was 46.0 ± 8.9, 63.3 ± 16.4, and 56.1 ± 16.0 years, respectively. The CKD stages of the patients in the CKD-ND group were as follows: two patients in CKD 1 (with more than moderately increased albuminuria), four in CKD 2 (with more than moderately increased albuminuria), six in CKD 3a, four in CKD 3b, and five in CKD 4. Patients in the CKD-ND and HD groups were older, had higher systolic blood pressures, and more comorbidities than the control group. None of the participants had any lung disease. Six participants had a history of cancer, but none reported cancer recurrence at the time of breath sampling. The mean eGFR of the normal, CKD-ND, and HD groups was 100.4 ± 12.0, 50.8 ± 26.9, and 4.4 ± 1.0 mL/min/1.73 m^2^, respectively. We divided the study participants into two groups for metabolomic analysis: a normal group (normal healthy controls) and a CKD group, which included patients with CKD-ND and HD.

### 3.2. Overview of Breath VOCs

A total of 324 VOCs with molecular weights ranging from 40 to 400 u were measured and classified into the following 16 VOC groups: volatile sulfur compounds (VSCs) 20, ketones 25, alcohols 43, halo-hydrocarbons 16, alkenes 13, alkyne 1, alkanes 68, terpenes 22, aromatics 26, acetates 25, acids 9, furans 3, steroids 1, aldehydes 12, nitrogen compounds 7, others 12, and siloxanes 21 (Appendix A). We excluded 21 VOCs belonging to the siloxane group from the analysis because they were assumed to be derived from methylpolysiloxane, which was coated onto the GC–MS column (Elite-5ms column, Perkin Elmer, Waltham, MA, USA) [16]. Before excluding them, we conducted an analysis that included the 21 VOCs from the siloxane group; however, no significant differences were observed. After excluding the 21 VOCs, 303 VOCs were included in the analysis to identify the breath markers of CKD.

### 3.3. Potential Breath Markers for CKD

We used an untargeted metabolomic strategy to identify marker breath VOCs associated with CKD and used a step-by-step approach to identify significant VOCs that could distinguish patients with CKD from normal controls. We analyzed 12 clinical variables to correct for confounding variables caused by the clinical characteristics or underlying comorbidities of the participants.

#### 3.3.1. Selection of Significant Breath VOCs and Clinical Variables

Among the six categorical variables, the χ^2^ test showed that diabetes, hypertension, dyslipidemia, and CHD were significantly higher in the CKD group compared with the control group (Table 1). Further, *t*-test analysis revealed significant differences in 58 breath VOCs and 2 clinical variables (age and SBP) between the CKD and normal groups (Appendix A).

In the univariate analysis, 36 VOCs and 2 clinical variables (age and SBP) were significantly associated with the CKD state: VSC 1, ketones 8, alcohols 2, halo-hydrocarbon 2, alkanes 8, terpenes 4, aromatics 5, acetate 1, aldehydes 2, and others 3 (10 VOC groups) (Table 2). In general, a heat plot analysis showed a greater and more distinct increase in breath VOCs as CKD progressed (Figure 2).

#### 3.3.2. Multivariate Analysis to Extract Prediction Model of Discriminating Patients with CKD from Normal Controls

We assessed the overall breath metabolite variation between groups using PCA. This analysis using thirty-six VOCs and two clinical variables from the univariate analysis showed the ability of TD–GCMS to detect changes in breath VOCs resulting from renal dysfunction by reducing many variables to their first- and second-order principal components (PC). PC1 was the first component that explained the largest degree of variation (57.9%) in the dataset, followed by PC2 (9.2%) and PC3 (8.0%) in the PCA model. PCA showed that the hemodialysis group could be separated from the normal group; further, the CKD-ND group was distributed between the HD and control groups (Figure 3).

To narrow down the variables, we selected continuous variables in the first step of the multivariate analysis through one-hundred-times-repeated training and test set analyses (Appendix A). The average accuracy of the 100 times multivariate models drawn from the training set was 71.4%. The selected significant continuous variables were age, 2-methyl-pentane (alkane), cyclohexanone (ketone), acetic acid ethyl ester (acetate), systolic blood pressure, and heptane (alkane). In Appendix A, we list these variables in order of frequency used in the 100 times multivariate models.

We integrated four significant categorical variables of comorbidities (diabetes, hypertension, dyslipidemia, and CHD) in the second step of multivariate modeling; however, there were no significant independent categorical variables. The final multivariate logistic regression model included three variables: age, 2-methyl-pentane (alkane), and cyclohexanone (ketone). The model was logit(p) = −12.92 + 0.18 × age + 2.10 × 2-methyl-pentane + 2.31 × cyclohexanone, where p is the probability of being a patient with CKD. This model which incorporated age (OR 0.18 (95% CI: 0.07–2.49), *p* = 0.013), 2-methyl-pentane (alkane) (OR 2.10 (95% CI: 0.94–2.24), *p* = 0.025), and cyclohexanone (ketone) (OR 2.31 (95% CI: 0.88–2.64), *p* = 0.008) showed high performance in identifying patients with CKD, with accuracy of 86.3% when applied to all 51 participants (Table 3). Receiver operating characteristic analysis revealed a high area under the curve (AUC) of 0.960 (Figure 4). The mean eGFR of patients in the CKD-ND group was 50.8 ± 26.9 mL/min/1.73 m^2^, which included many patients with CKD in the early stage, but the final model discriminated patients with CKD from normal controls with high accuracy. Figure 5 shows the significant changes in cyclohexanone and 2-methyl-pentane levels in the normal, CKD-ND, and HD groups (before and after hemodialysis).

## 4. Discussion

In this untargeted breath analysis study using TD–GCMS, we identified and measured 303 VOCs from human breath, excluding 21 contaminant siloxane compounds. We showed that breath VOCs are a good marker for discriminating between patients with CKD and healthy controls, even after correcting for many clinical confounding variables. Thirty-six breath VOCs were significantly different between patients with CKD and normal controls in the univariate analysis. We also derived a final multivariate model that incorporated age and breath 2-methyl-pentane and cyclohexanone, producing a high accuracy of 86.3% in predicting CKD.

Considering the accumulation of putative uremic retention metabolites as a result of progressive kidney dysfunction, it is reasonable to assume that CKD leads to a characteristic chemical profile of breath VOCs. The list of uremic retention solutes is constantly evolving as new compounds are discovered [5]. In this context, an untargeted approach to breath analysis is more appropriate than focusing on specific breath VOCs to identify CKD breath markers. To our knowledge, this is the first study to provide a general overview of breath VOCs in patients with CKD as well as end stage renal disease TD–GCMS is a powerful analytical technique that is often used to analyze VOCs in air and human breath. One of the main advantages of TD–GCMS is its ability to separate and identify a wide range of VOCs. TD–GCMS is considered one of the most sensitive and selective techniques available [17]. We controlled for the effect of VOCs in the room air by subtracting the VOC value of the room air from the VOC values of the participants. In this study, we showed that 36 breath VOCs were significantly related to CKD in the univariate analysis, and the heat plot of these VOCs revealed that many of these breath VOCs increased with decreasing renal function. Changes in breath VOCs associated with decreased renal function were also confirmed using PCA. As renal dysfunction progressed from the normal group to the hemodialysis group, the spectrum of breath VOC patterns shifted accordingly. The CKD-ND group included patients with a wide range of eGFR values, as shown in Figure 5a. The lower the eGFR, the higher the breath VOCs, which reflects the accumulation of uremic retention solutes, resulting in overlap between the HD and CKD-ND groups in Figure 3. Six patients in the CKD-ND group were CKD 1–2 stage, and this may have affected the overlap between the normal and CKD-ND groups in the PCA analysis. Nevertheless, the final multivariate model showed a high accuracy of 86%.

In previous animal and human studies, cyclohexanone was used as a solvent in the production of extracorporeal circuits and intravenous bags and was considered one of the contaminants from extracorporeal materials, for example, during hemodialysis. The increase in breath cyclohexanone levels after hemodialysis (Figure 5c) was in good agreement with the results of previous reports [18,19]. However, in our study, cyclohexanone was detected in the breath of normal participants and non-dialysis patients with CKD and hemodialysis and showed an increasing trend with decreasing renal function, with the highest level in hemodialysis patients after adjusting for room-air cyclohexanone. Cyclohexanone (CAS No. 108-94-1), a six-carbon cyclic compound with a ketone group, is an organic compound with the chemical formula (CH_2_)_5_CO that is miscible with water. This colorless liquid had a sweet and pungent smell reminiscent of acetone. One of the most common uses of cyclohexanone is in the production of nylon, which is used in various end-use industries, including automotive, construction, consumer goods, and electronics [20]. Cyclohexanone is a synthetic compound that is not produced by the human body. However, limited information is available regarding the role of cyclohexanone in human metabolism. Cyclohexanone is well absorbed through the skin, respiratory tract, and alimentary tract and metabolized to cyclohexanol, which is conjugated with glucuronic acid and excreted mainly in the urine [21,22,23]. However, cyclohexanone toxicity has rarely been documented in humans or experimental animals [24,25]. Ong et al. studied occupational exposure to cyclohexanone by analyzing the breath and urine of 59 workers [26]. The possible source of breath cyclohexanone detected in normal controls and non-dialysis patients with CKD in this study is not certain and may be exposure through the use of products or inhalation of gases that contain cyclohexanone-related chemicals. However, a decrease in kidney function resulted in significantly increased levels of breath cyclohexanone. Cyclohexanone was retained after hemodialysis, as shown in Figure 5c, because it is a soluble chemical and may be relatively slowly removed from the blood by exhalation in hemodialysis patients in whom urinary excretion of its metabolites is almost absent. Mochalski et al. reported that uremic breath is affected by contaminants from extracorporeal circuits, and cyclohexanone, one of those contaminants, can play a role as a uremic toxin because it is retained after hemodialysis [19].

2-methyl-pentane (C6H14, CAS NO. 107-83-5), also known as isohexane (an isomer of hexane), is a branched-chain alkane (subclassification of hydrocarbons) with a gasoline-like odor that floats on water. It is a colorless, flammable liquid commonly used as a solvent in industry. Solvents containing hexane are primarily used to extract vegetable oils from crops such as soybeans. These solvents are used as cleaning agents in the printing, textile, furniture, and shoe-making industries [27]. It is also used in several consumer products, such as gasoline, quick-drying glues, and rubber cement [28]. 2-methyl-pentane is distributed throughout the body in the blood and metabolized by mixed-function oxidases in the liver into a number of metabolites [29]. It has been detected in the feces, breath, saliva, and blood of healthy humans [30,31]. Further, it has also been reported that 2-methyl-pentane increases the breathing of preschool asthmatic children [32]. 2-methyl-pentane was detected in the blood and breath of hemodialysis patients and was classified as a contaminant from dialyzers and bloodlines [19]. We do not know the origin and metabolic fate of the 2-methyl-pentane detected in this study. It was rarely detected in the breath of normal controls, except in three participants, but was increased in the breath of non-dialysis patients with CKD, was higher in those of hemodialysis patients, and was not removed after hemodialysis (Figure 5d).

This study had several limitations. The results were obtained from the analysis of VOCs in a small number of breath samples in a hospital environment. Therefore, these results require further validation. The mean age of patients with CKD was higher than that of normal participants. We did not measure the blood levels of the corresponding breath VOCs, and hence the significance of VOCs needs to be explored in the context of internal and external metabolism. Unexpectedly, some chemicals exposed to the environment were measured through exhalation, and they accumulated as kidney function decreased, even if a person did not receive hemodialysis. This study and earlier studies have shown that as kidney function deteriorates, various uremic retention metabolites accumulate in the body, which is reflected in increased breath VOCs. Further research is needed to fully understand the relationship between CKD and breath VOCs and to determine whether breath VOCs can be used as a diagnostic tool for CKD. However, the presence of certain VOCs in the breath of patients with CKD may provide valuable information regarding disease progression and treatment efficacy.

## 5. Conclusions

This study showed that renal dysfunction induces a characteristic profile of breath VOCs. Knowledge of these specific breath VOC profiles facilitates their use as potential biomarkers and may enable the transition from blood test to non-invasive breath analysis in screening tests for CKD.

## Figures and Tables

**Figure 1 metabolites-13-00837-f001:**
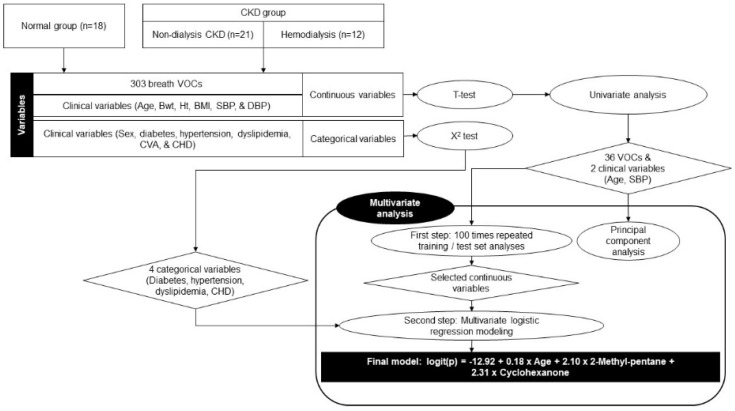
Flow chart of statistical analyses. Abbreviations: CKD (chronic kidney disease); VOC (volatile organic compound); CVA (cerebrovascular accident); CHD (coronary heart disease); BMI (body mass index); SBP (systolic blood pressure); DBP (diastolic blood pressure).

**Figure 2 metabolites-13-00837-f002:**
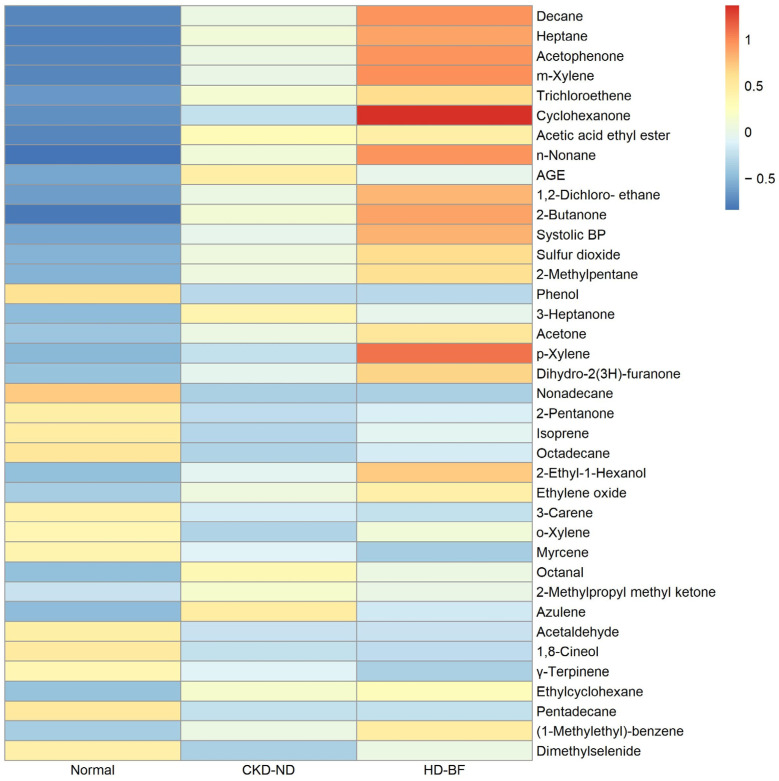
Heat plot of thirty-six breath volatile organic compounds and two clinical variables related to chronic kidney disease in the univariate analysis. (Order is the same as that in Table 2.) Abbreviations: CKD-ND (non-dialysis CKD); HD-BF (before hemodialysis).

**Figure 3 metabolites-13-00837-f003:**
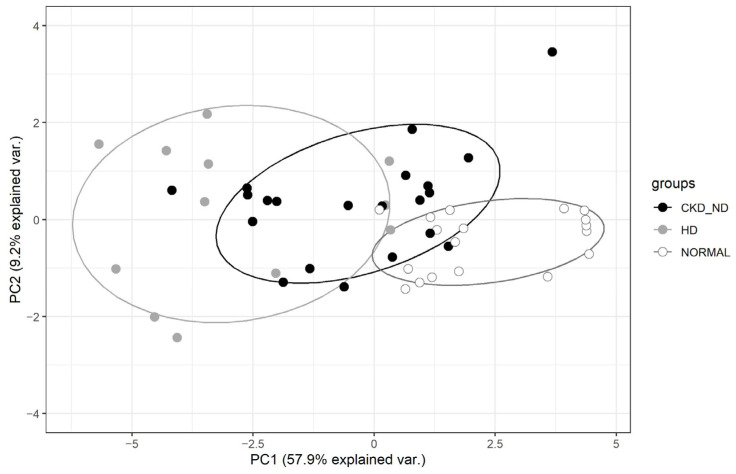
Multivariate analysis. Principal component analysis (PCA) of metabolomic profiles of breath samples from the control (NORMAL), non-dialysis chronic kidney disease (CKD_ND), and hemodialysis (HD) groups using 36 significant volatile organic compounds and 2 significant clinical variables in univariate analysis (as shown in Table 2). PC1 and PC2 indicate the first and second component that explained the largest degree of variation in the dataset of the PCA model, 57.9% and 9.2%, respectively.

**Figure 4 metabolites-13-00837-f004:**
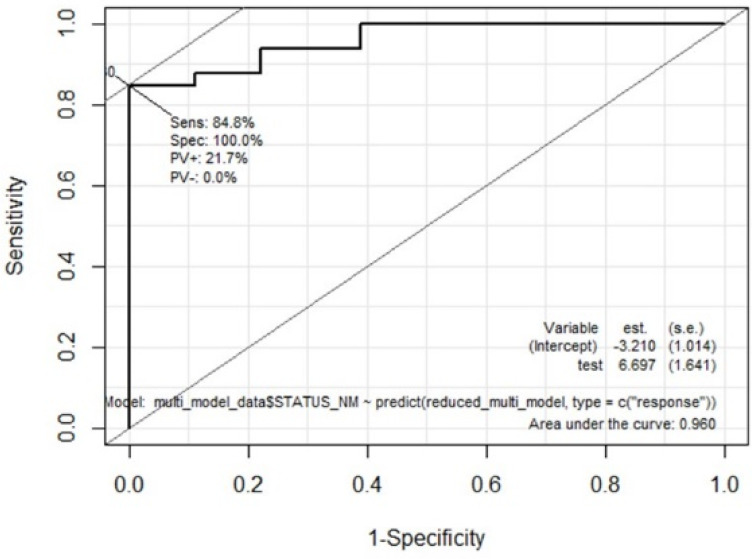
Receiver operating characteristic analysis of the final multivariate model incorporating age and levels of 2-methyl-pentane and cyclohexanone. Abbreviations: AUC (area under the curve).

**Figure 5 metabolites-13-00837-f005:**
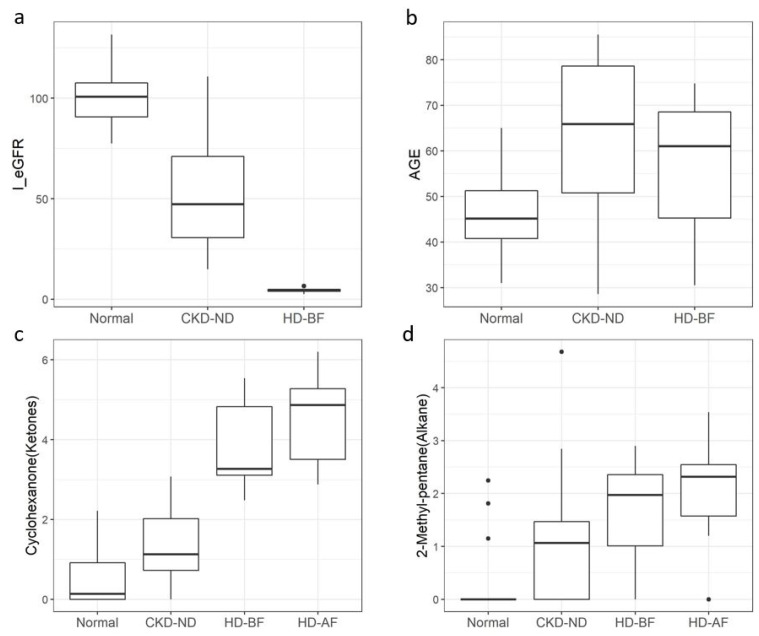
Estimated glomerular filtration rate (GFR) and the three features that most contributed to the final model. (**a**) distribution of eGFR (CKD-EPI) of the normal control group versus the CKD groups (CKD_ND and HD). (**b**) age. (**c**) cyclohexanone (ketone). (**d**) 2-methyl-pentane (alkane). Boxes show from 25 to 75 percentile distribution, whereas whiskers indicate 95% confidence interval. The *y*-axis of breath volatile organic compounds (VOCs) in c and d represents the scaled-down VOC values according to the following modification: SQRT (breath VOC value-background air VOC value)/1000. Abbreviations: CKD-ND (non-dialysis CKD); HD-BF (before hemodialysis); HD-AF (after hemodialysis).

**Table 1 metabolites-13-00837-t001:** Characteristics of participants.

Total *n* = 51	Normal Healthy Control Group (*n* = 18)	CKD Group		
		Non-Dialysis CKD Patients (*n* = 21)	Hemodialysis Patients (*n* = 12)	*p*-Value
Age	46.0 ± 8.9 (31.0–65.0)	63.3 ± 16.4 (28.6–85.5)	56.1 ± 16.0 (30.5–74.8)	0.0001
Sex (M:F)	10:8	13:8	6:6	0.8893
Physical Examination (mean ± SD)				
Body weight	67.5 ± 14.6	71.9 ± 14.5	59.2 ± 10.3	0.962
Height	168.0 ± 9.5	163.6 ± 10.5	162.5 ± 10.0	0.119
Body Mass Index (BMI)	23.7 ± 3.2	26.7 ± 4.1	22.3 ± 2.6	0.230
Systolic BP (SBP)	122.4 ± 13.7	133.1 ± 13.9	148.3 ± 21.7	0.003
Diastolic BP (DBP)	76.3 ± 9.7	74.9 ± 10.2	78.3 ± 14.0	0.955
Comorbidity (number (%))				
Diabetes	0 (0%)	10 (47.6%)	7 (58.3%)	0.0000
Hypertension	0 (0%)	17 (81.0%)	11 (91.7%)	0.0000
Dyslipidemia	0 (0%)	17 (81.0%)	3 (25.0%)	0.0000
Cerebrovascular accident (CVA)	0 (0%)	2 (9.5%)	1 (8.3%)	0.0995
Coronary heart disease (CHD)	0 (0%)	1 (4.8%)	7 (58.3%)	0.0053
Lung disease	0 (0%)	0 (0.0%)	0 (0.0%)	
History of cancer	1 (5.6%)	5 (23.8%)	0 (0.0%)	
Lab data (mean ± SD)				
BUN	11.0 ± 3.9	23.6 ± 8.6	62.4 ± 20.0	
Serum creatinine	0.80 ± 0.16	1.56 ± 0.55	10.49 ± 1.87	
eGFR (CKD-EPI)	100.4 ± 12.0	50.8 ± 26.9	4.4 ± 1.0	
Hemoglobin	14.2 ± 1.3	13.2 ± 2.3	10.8 ± 1.0	
Glucose	101.4 ± 10.0	118.2 ± 26.9	150.8 ± 55.3	
Calcium	9.6 ± 0.4	9.4 ± 0.8	8.9 ± 0.5	
Phosphorus	3.6 ± 0.4	3.6 ± 0.5	5.1 ± 1.0	
Uric acid	4.7 ± 1.7	7.1 ± 1.5	6.3 ± 1.1	
Total cholesterol	201.8 ± 18.6	161.3 ± 48.8	107.8 ± 17.9	
Triglycerides	111.8 ± 56.8	125.1 ± 51.3	100.0 ± 55.3	
HDL-cholesterol	63.5 ± 13.4	48.4 ± 9.3	47.2 ± 7.9	
LDL-cholesterol	122.7 ± 19.4	90.6 ± 36.4	55.0 ± 12.3	
Protein	7.5 ± 0.5	7.1 ± 0.6	6.6 ± 0.4	
Albumin	4.7 ± 0.4	4.2 ± 0.6	3.9 ± 0.3	

**Table 2 metabolites-13-00837-t002:** Thirty-six breath VOCs and two clinical variables that were significant in CKD (non-dialysis CKD + Hemodialysis) compared to normal healthy control groups by using univariate analysis in the order of *p*-value. (OR: odds ratio).

Name of VOC or Clinical Variable	VOC Group	OR	95% CI	*p*_Value	DEP_VAR
Decane	Alkane	2.79	1.67–5.44	0.0005	A_152
Heptane	Alkane	5.85	2.47–20.09	0.0006	A_134
Acetophenone	Ketones	10.04	3.15–50.01	0.0008	A_35
m-Xylene	Aromatics	2.27	1.5–4.13	0.0010	A_222
Trichloroethene	Halo-Hydrocarbons	4.00	1.88–10.48	0.0013	A_100
Cyclohexanone	Ketones	3.60	1.83–9.04	0.0014	A_31
Acetic acid ethyl ester	Acetate	3.63	1.86–9.5	0.0014	A_239
n-Nonane	Alkane	2.69	1.52–5.68	0.0025	A_147
Age		1.08	1.03–1.14	0.0035	AGE
1,2-Dichloro- ethane	Halo-Hydrocarbons	16.48	3.27–169.2	0.0043	A_93
2-Butanone	Ketones	2.07	1.29–3.77	0.0069	A_23
Systolic BP		1.07	1.02–1.13	0.0085	PE_SBP
Sulfur dioxide	VSC	3.03	1.45–7.84	0.0085	A_4
2-Methyl pentane	Alkane	2.92	1.42–7.54	0.0099	A_130
Phenol	Alcohol	0.78	0.61–0.91	0.0105	A_62
3-Heptanone	Ketones	5.56	1.63–26.17	0.0135	A_25
Acetone	Ketones	1.05	1.01–1.11	0.0159	A_21
p-Xylene	Aromatics	2.14	1.26–4.54	0.0166	A_219
Dihydro-2(3H)-furanone	Ketones	2.59	1.26–6.17	0.0171	A_27
Nonadecane	Alkane	0.80	0.62–0.91	0.0175	A_178
2-Pentanone	Ketones	0.72	0.53–0.92	0.0176	A_28
Isoprene	Terpenes	0.96	0.91–0.99	0.0188	A_187
Octadecane	Alkane	0.51	0.26–0.82	0.0241	A_177
2-Ethyl-1-Hexanol	Alcohol	2.64	1.34–8.7	0.0264	A_67
Ethylene oxide	Other	1.49	1.06–2.17	0.0269	A_294
3-Carene	Terpenes	0.65	0.42–0.91	0.0271	A_197
o-Xylene	Aromatics	0.79	0.63–0.96	0.0282	A_221
Myrcene	Terpenes	0.73	0.54–0.93	0.0293	A_196
Octanal	Aldehyde	3.03	1.31–10.64	0.0309	A_281
2-Methylpropyl methyl ketone	Ketones	2.74	1.14–7.4	0.0327	A_33
Azulene	Aromatics	4.71	1.52–37.23	0.0351	A_228
Acetaldehyde	Aldehyde	0.85	0.7–0.97	0.0353	A_273
1,8-Cineol	Other	0.85	0.69–0.96	0.0378	A_297
γ-Terpinene	Terpenes	0.72	0.5–0.95	0.0387	A_192
Ethyl cyclohexane	Alkane	5.99	1.48–61.03	0.0444	A_139
Pentadecane	Alkane	0.71	0.45–0.91	0.0450	A_169
(1-Methylethyl)-benzene	Aromatics	2.41	1.11–6.59	0.0468	A_223
Dimethyl selenide	Other	0.69	0.46–0.98	0.0481	A_285

**Table 3 metabolites-13-00837-t003:** Final multivariate logistic regression model with accuracy obtained by applying the final model to 51 participants (Akaike information criterion (AIC) = 30.3).

DEP_VAR	OR	2.50%	97.50%	*p*-Value
(Intercept)	−12.92	4.80	−2.69	0.007
Age	0.18	0.07	2.49	0.013
2-Methyl-pentane (A_130)	2.10	0.94	2.24	0.025
Cyclohexanone (A_31)	2.31	0.88	2.64	0.008
**STATUS_NM**	**0**	**1**		
0	15	3		
1	4	29	Accuracy = 86.3%	

## Data Availability

The datasets of the current study are available from the corresponding author on reasonable request. The data are not publicly available due to privacy.

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
