# Peer review of "Exploration of Potential Breath Biomarkers of Chronic Kidney Disease through Thermal Desorption–Gas Chromatography/Mass Spectrometry"

_metabolites, 2023, doi:10.3390/metabo13070837_

Round 1

Reviewer 1 Report

line 38 needs to be fixed from "with odds ratios (95% confidence interval) P-values of 0.18 38 (0.07-2.49) 0.013; 2.10 (0.94-2.24) 0.025; and 2.31 (0.88–2.64) 0.008, respectively. " to"with odds ratios of  0.18 38 (95% CI, 0.07-2.49, p=0.013); 2.10 (0.94-2.24, p= 0.025); and 2.31 (0.88–2.64, p=0.008), respectively. "

in abstract use acronym for breath volatile organic compounds.

only 47% of individuals in non-dialysis CKD have diabetes. having diabetes and CKD together may add more load to the VOCs detected. authors then should separate non-dialysis non-diabetes CKD (n=11) as a separate group. 5 individuals have cancer how can the author justify that the VOCs observed are due to CKD? One individual in the normal group has cancer, on which ground this individual was considered normal. re-analyze the data after removing cancer patients? CKD pt with and without diabetes should be separately analyzed.

figures are of very low quality, if readers cannot read of the figures what purpose these figures serve?

define Bwt in table 1

rephrase and simplify the concluding sentence in abstract (lines 39-40)  "Hence, renal dysfunction-associated characteristic profiles of breath volatile organic substances can be used as non-invasive markers to screen for chronic kidney disease." 

There are only 51 samples, the authors performed 100 repeated samplings to create test and train datasets, with such low n, how many samples were repeatedly selected? Also, the authors need to discuss the overfitting of the model due to some samples repeatedly being selected in the datasets. improve quality of Figure 1.

there is a significant overlap between CKD-ND pt are in the HD group in figure 3, the authors need to discuss this. can authors identify diabetes CKD and non diabetes CKD in this figure as well.

it is not clear if figure 4 is of a single VOC or a combination of VOCs. how was the data for VOCs was combined. what is the purpose of this ROC analysis.

discussion lists a lot on individual VOCs, pl discuss your results in light of existing similar human studies. mixing animal studies with human studies should be avoided.

conclusions reads like the authors themselves are not convinced of their own results.

the authors should discuss table S2. infact a bubble chart can be used to show the information, bubbles representing the column 1 value and color density representing p-value. 

it is not clear if in all the 100 boot strapping, the regression formula remained same or changed.

quality of English is ok. 

Author Response

  1. line 38 needs to be fixed from "with odds ratios (95% confidence interval) P-values of 0.18 38 (0.07-2.49) 0.013; 2.10 (0.94-2.24) 0.025; and 2.31 (0.88–2.64) 0.008, respectively. " to"with odds ratios of  0.18 38 (95% CI, 0.07-2.49, p=0.013); 2.10 (0.94-2.24, p= 0.025); and 2.31 (0.88–2.64, p=0.008), respectively. "
    1. Reply:  Thank you for your kind comment. We revised it as commented.
  2. in abstract use acronym for breath volatile organic compounds.
    1. Reply: We revised it as commented.
  3. only 47% of individuals in non-dialysis CKD have diabetes. having diabetes and CKD together may add more load to the VOCs detected. authors then should separate non-dialysis non-diabetes CKD (n=11) as a separate group. 5 individuals have cancer how can the author justify that the VOCs observed are due to CKD? One individual in the normal group has cancer, on which ground this individual was considered normal. re-analyze the data after removing cancer patients? CKD pt with and without diabetes should be separately analyzed.
    1. Reply: Thanks for the great comments. The authors were also concerned about the effect of diabetes on breath VOCs in patients with chronic kidney disease while analyzing the breath VOC data, but DM was not an independent  variable of significance in the final multivariate analysis. This was described in the submitted manuscript as follows;
      1. In lines 266 to 268, “We integrated four significant categorical variables of comorbidities (diabetes, hypertension, dyslipidemia, and CHD) in the second step of multivariate modeling; however, there were no significant independent categorical variables. “
  4. figures are of very low quality, if readers cannot read of the figures what purpose these figures serve?
    1. Reply: We totally agree with you and revised the figure 1 and 5 accordingly.
  5. define Bwt in table 1
    1. Reply:  Bwt was converted to Body weight.
  6. rephrase and simplify the concluding sentence in abstract (lines 39-40)  "Hence, renal dysfunction-associated characteristic profiles of breath volatile organic substances can be used as non-invasive markers to screen for chronic kidney disease."
    1. Reply: We agree with you. The concluding sentence in the abstract was revised.
  7. There are only 51 samples, the authors performed 100 repeated samplings to create test and train datasets, with such low n, how many samples were repeatedly selected? Also, the authors need to discuss the overfitting of the model due to some samples repeatedly being selected in the datasets.
    1. Reply: Thank you for the detailed paper review. As pointed out, the total number of participants in this study was small at 51. The authors tried to find an optimal analysis method to compensate for this. The answer to the reviewer's question "how many samples were repeatedly selected?" can be found in table S3. If the numbers of the four cases in Table S3 are added, that is the number of subjects included in the test set. In repeated sampling of 100 times, 33% of the subjects, that is, 11-13 subjects, were selected as the test set each time. This is described in an existing paper as follows.
      1. In lines 168 to 173, “Next, multivariate analysis was performed in two steps. First, significant continuous variables were selected from 100 repeated training and test sets. Briefly, 51 participants were randomly sampled into the training set (67%) and the test set (33%), and a multivariate model was constructed with the training dataset; further, the accuracy was derived by applying this model to the corresponding test set. This process was repeated a hundred times.”
    2. Reply: Overfitting of the final model, as pointed out by the reviewer, is an important issue. The authors checked the AIC, Akaike information criterion,  and the p value of each variable in the process of constructing the final multivariate model to avoid overfitting, which can be seen in Table 5. AIC is designed to avoid overfitting by penalizing models which use more independent variables (parameters). AIC is most often used to compare the relative goodness-of-fit among different models under consideration and to then choose the model that best fits the data.  AIC is most useful when working with a small data set. In the submitted manuscript, this is described as follows;
      1. In lines 175-179, “In the second step, the final multivariate logistic regression model was built by integrating the selected continuous variables in the first step and four significant categorical variables selected using the χ2 test. We used stepwise regression with forward selection and backward elimination to obtain a model that minimized the Akaike information criterion (AIC) while avoiding overfitting.”
  8. there is a significant overlap between CKD-ND pt are in the HD group in figure 3, the authors need to discuss this. can authors identify diabetes CKD and non diabetes CKD in this figure as well.
    1. Reply:  Thank you for the valuable comment. The CKD-ND group included patients with wide range eGFR as shown in figure 5a. The lower the eGFR, the higher breath VOCs, which reflects the accumulation of uremic retention solutes, resulting in overlap between the HD and CKD-ND groups in figure 3. The authors have added this to the discussion (in line from 327 to 330 in the revised manuscript). We are sorry for not being able to identify the diabetic states in figure 3.  As mentioned above, diabetes was not an independent  variable of significance in the final multivariate analysis.
  9. it is not clear if figure 4 is of a single VOC or a combination of VOCs. how was the data for VOCs was combined. what is the purpose of this ROC analysis.
    1. Reply: Figure 4, the heat map, represents a single VOC. ROC analysis was done to demonstrate the high performance of the final multivariate logistic regression model  in identifying patients with CKD.
  10. discussion lists a lot on individual VOCs, pl discuss your results in light of existing similar human studies. mixing animal studies with human studies should be avoided.
    1. Reply: Thank you for your comment. Descriptions of animal study have been removed.
  11. conclusions reads like the authors themselves are not convinced of their own results.
    1. Reply: The authors agree with you. We rewrite the conclusion.
  12. the authors should discuss table S2. infact a bubble chart can be used to show the information, bubbles representing the column 1 value and color density representing p-value.
    1. Reply: Table S2 is the result of the t-test, and was the initial step in the process of selecting significant VOCs among over 300 breath VOCs. Bubble charts are used to show three types of information at once, and the result of the t test provides one information, the p value. What values can be selected for the x-axis and y-axis to configure the bubble chart?  We are sorry, but we think that it is difficult to display the results of the t test in a bubble chart.
  13. it is not clear if in all the 100 boot strapping, the regression formula remained same or changed.
    1. Reply:  Of course, each multivariate model was constructed for each boot strapping.  The 'model' column on the far right of Table S 3 describes the variables used in each multivariate model. This is described in the submitted manuscript as follows.
      1. In lines 168 to 174, “First, significant continuous variables were selected from 100 repeated training and test sets. Briefly, 51 participants were randomly sampled into the training set (67%) and the test set (33%), and a multivariate model was constructed with the training dataset; further, the accuracy was derived by applying this model to the corresponding test set. This process was repeated hundred times and the variables used in each multivariate model were listed based on the number of times they were used.”

Reviewer 2 Report

Dear author. I am pleased to read the manuscript. I find the study very interesting and relevant. 

I have only few issues that have to be resolved prior to publishing. I have noted that you have selected three groups, CKD, CKD+HD and controls. HD group is not really interesting from diagnostic point of view but it is very useful for identifying certain markers. CKD+HD group is the one that would benefit the most from new diagnostic method. What I miss is the description or better said staging of the group. You have provided GFR data but I believe the detailed structure of the group is extremely important for the interpretation of the final models. Additionally, I would like to see a little bit more emphasis on the differentiation between CKD_ND and control group since this is the one that will probably be used in the future. Otherwise, congratulations on the study!

Author Response

We really appreciate your valuable comment. The CKD stages of the patients in the CKD-ND group are as follows; 2 patients in CKD 1 (with more than moderately increased albuminuria), 4 in CKD 2 (with more than moderately increased albuminuria), 6 in CKD 3a, 4 in CKD 3b and 5 in CKD 4.  This information has been included in the revised manuscript (please refer to the line 197-199 in the revised manuscript). Six patients in the CKD-ND group were CKD 1-2 stage, and this may have affected the overlap between the normal and CKD-ND groups in the PCA analysis, as shown in figure 3. Nevertheless, the final multivariate model showed a high accuracy of 86%. In the future study, we try to improve study design by supplementing this point. Thank you again for your comment.

Round 2

Reviewer 1 Report

The authors addressed all my comments